# On-Treatment Changes in FIB-4 and 1-Year FIB-4 Values Help Identify Patients with Chronic Hepatitis B Receiving Entecavir Therapy Who Have the Lowest Risk of Hepatocellular Carcinoma

**DOI:** 10.3390/cancers12051177

**Published:** 2020-05-07

**Authors:** Hung-Wei Wang, Hsueh-Chou Lai, Tsung-Hui Hu, Wen-Pang Su, Sheng-Nan Lu, Chia-Hsin Lin, Chao-Hung Hung, Po-Heng Chuang, Jing-Houng Wang, Mei-Hsuan Lee, Chien-Hung Chen, Cheng-Yuan Peng

**Affiliations:** 1Center for Digestive Medicine, Department of Internal Medicine, China Medical University Hospital, Taichung 404, Taiwan; sdqw190@gmail.com (H.-W.W.); t674233@ms54.hinet.net (H.-C.L.); dadun2022@yahoo.com.tw (W.-P.S.); sk19850121@gmail.com (C.-H.L.); poheng2@yahoo.com.tw (P.-H.C.); 2School of Chinese Medicine, China Medical University, Taichung 404, Taiwan; 3Division of Hepatogastroenterology, Department of Internal Medicine, Kaohsiung Chang Gung Memorial Hospital and Chang Gung University College of Medicine, Kaohsiung 833, Taiwan; dr.hu@msa.hinet.net (T.-H.H.); juten@ms17.hinet.net (S.-N.L.); chh4366@yahoo.com.tw (C.-H.H.); jinghoung2001@yahoo.com.tw (J.-H.W.); 4Institute of Clinical Medicine, National Yang-Ming University, Taipei 112, Taiwan; meihlee@ntu.edu.tw; 5School of Medicine, China Medical University, Taichung 404, Taiwan

**Keywords:** chronic hepatitis B, entecavir, hepatocellular carcinoma, fibrosis-4 (FIB-4), noninvasive fibrosis index

## Abstract

Noninvasive fibrosis indices can help stratify the risk of hepatocellular carcinoma (HCC) in patients with chronic hepatitis B (CHB) receiving nucleos(t)ide analogue (NA) therapy. We investigated the predictive performance of on-treatment changes in FIB-4 (△FIB-4) and 1-year FIB-4 values (FIB-4 12M) for HCC risk in patients with CHB receiving entecavir therapy. We included 1325 NA-naïve patients with CHB treated with entecavir, retrospectively, from January 2007 to August 2012. A combination of △FIB-4 and FIB-4 12M was used to stratify the cumulative risk of HCC into three subgroups each in the noncirrhotic and cirrhotic subgroups with *p* < 0.0001 by using the log-rank test (noncirrhotic: the highest risk (*n* = 88): FIB-4 12M ≥ 1.58/△FIB-4 ≥ 0 (hazard ratio (HR): 40.35; 95% confidence interval (CI): 5.107–318.7; *p* <0.0001) and cirrhotic: the highest risk (*n* = 89): FIB-4 12M ≥2.88/△FIB-4 ≥0 (HR: 9.576; 95% CI: 5.033–18.22; *p* < 0.0001)). Patients with noncirrhotic CHB treated with entecavir who had a FIB-4 12M < 1.58 or FIB-4 12M ≥ 1.58/△FIB-4 < 0 exhibited the lowest 5-year HCC risk (0.6%). A combination of on-treatment changes in FIB-4 and 1-year FIB-4 values may help identify patients with CHB receiving entecavir therapy with the lowest risk of HCC.

## 1. Introduction

Liver cirrhosis and hepatocellular carcinoma (HCC) are the major complications of chronic hepatitis B (CHB) [1]. Long-term nucleos(t)ide analogue (NA) therapy leads to the regression of liver fibrosis and cirrhosis and reduces the incidence of HCC [2,3,4,5]. Unfortunately, the risk of HCC occurrence remains, although in a recent large cohort of 1951 Caucasian patients with CHB under entecavir or tenofovir treatment, the 8-year survival rate was found to be similar to that for the general population [6,7,8]. The severity of liver fibrosis has been known to be a crucial risk factor for HCC development. Many noninvasive indices or modalities have been utilised instead of liver biopsy to predict liver fibrosis status [9,10,11]. The aminotransferase-to-platelet ratio index (APRI, Equation (1)) and fibrosis-4 (FIB-4, Equation (2)) index are popular noninvasive indices to predict liver fibrosis in chronic viral hepatitis [12,13,14]. Furthermore, baseline FIB-4 has a higher predictive performance for HCC than other indices in patients with CHB [15,16,17]. An Asian study reported that patients with noncirrhotic CHB receiving long-term NA therapy who had a FIB-4 < 1.29 at baseline had the lowest risk for HCC [18]. Recent studies in Caucasian patients with CHB revealed that a low platelet count at baseline and liver stiffness measurement ≥ 12 kPa at Year 5 was an independent predictor of HCC development before and after 5 years of entecavir or tenofovir treatment, respectively [19,20,21]. However, limited reports have addressed how the on-treatment dynamic change or on-treatment value of noninvasive indices affects HCC development. We therefore conducted this study to investigate the risk of HCC according to on-treatment changes in FIB-4 and 1-year FIB-4 values in 1325 NA-naïve patients with CHB treated with entecavir.

## 2. Results

### 2.1. Baseline and On-Treatment Characteristics

The median age of the study group at baseline was 50 ± 17 years. Of the patients, 963 (72.7%) were male, 481 (36.3%) were cirrhotic, and 158 (11.9%) were diabetic. A total of 105 patients (7.9%) developed HCC during a median treatment duration or follow-up period of 4.1 years. The laboratory variables and noninvasive fibrosis indices at baseline and after 1 year of treatment are presented in Table 1. We compared the characteristics of the patients with HCC (*n* = 105) with those without HCC (*n* = 1220). Older age; higher percentages of diabetes mellitus (DM) and cirrhosis; lower serum albumin, aspartate aminotransferase (AST), alanine aminotransferase (ALT), hepatitis B virus (HBV) deoxyribonucleic acid (DNA), and platelet counts; and higher international normalised ratios (INRs), alpha-fetoprotein (AFP), and AFP 12M levels were observed in the patients with HCC (Table 1). Most of the patients with and without HCC had achieved a virological response (VR) after 1 year of entecavir therapy (93.3% and 87%, respectively).

### 2.2. APRI, FIB-4, and On-Treatment Changes

Table 1 shows the APRI and FIB-4 values at baseline and after 1 year of treatment and presents the on-treatment changes (12M–0M) in all the patients and in the subgroups with and without HCC. Those values at baseline and after 1 year of treatment for the cirrhotic subgroup were 1.22 ± 1.63 and 0.69 ± 0.57 for APRI, and 3.22 ± 3.57 and 2.61 ± 2.24 for FIB-4, respectively (Appendix A). The optimal values of APRI, APRI 12M, FIB-4, and FIB-4 12M to differentiate HCC risk were 0.90, 0.53, 2.53, and 2.56, respectively. The sensitivity, specificity, positive predictive value (PPV), negative predictive value (NPV), and accuracy according to the optimal cutoff values of APRI and FIB-4 indices at baseline or after 1 year of treatment in predicting HCC risk are shown in Appendix A, and their areas under the receiver operating characteristic (AUROCs) curve were compared (Appendix A). The median APRI 12M, FIB-4, and FIB-4 12M values and the proportions of patients with APRI ≥ 0.90, APRI 12M ≥ 0.53, △APRI ≥ 0, FIB-4 ≥ 2.53, FIB-4 12M ≥ 2.56, and △FIB-4 ≥ 0 were significantly different between the patients with HCC and those without HCC (Table 1).

### 2.3. Baseline and On-Treatment Factors Associated with HCC Occurrence

In all of the patients, age, cirrhosis status, DM, albumin, AST, ALT, platelets, AFP 12M, APRI ≥ 0.90, APRI 12M ≥ 0.53, △APRI ≥ 0, FIB-4 ≥ 2.53, FIB-4 12M ≥ 2.56, and △FIB-4 ≥ 0 showed significant associations with HCC according to univariate Cox regression analysis (Table 2). By employing multivariate Cox regression analysis (FIB-4 based model without the inclusion of the cirrhosis factor), DM, AFP 12M, FIB-4 12M ≥ 2.56, and △FIB-4 ≥ 0 were determined to be independent predictors of HCC (Table 3 and Appendix A). In the noncirrhotic subgroup, FIB-4 12M ≥ 1.58 and △FIB-4 ≥ 0 were independent predictors of HCC, whereas in the cirrhotic subgroup, sex, DM, AFP 12M, FIB-4 12M ≥ 2.88, and △FIB-4 ≥ 0 were independent predictors of HCC. These results were obtained through multivariate Cox regression analysis (Table 3). The results of univariate Cox regression analysis are shown in Appendix A.

### 2.4. Cumulative Incidence of HCC Stratified by 1-Year FIB-4 Values or On-Treatment Changes in FIB-4

The results of our multivariate Cox regression analysis revealed that FIB-4 12M and △FIB-4 were independent predictors for HCC in all of the patients and subgroups with and without cirrhosis. In Figure 1, both FIB-4 12M and △FIB-4 are shown to stratify the risk of HCC (log-rank test of both *p* < 0.0001). The 5-year cumulative incidences of HCC stratified by FIB-4 12M were 27.2% vs. 2.7% for all of the patients with FIB-4 12M ≥ 2.56 vs. < 2.56; 5.9% vs. 0.2% for noncirrhotic patients with FIB-4 12M ≥ 1.58 vs. < 1.58; and 33.5% vs. 8% for the cirrhotic patients with FIB-4 12M ≥ 2.88 vs. < 2.88.

### 2.5. Comparison of Baseline Characteristics among the Subgroups Stratified by On-Treatment Changes in FIB-4

According to the on-treatment changes in FIB-4, we categorised the treatment-naïve patients with CHB into subgroups with △FIB-4 < 0 (*n* = 1002) or △FIB-4 ≥ 0 (*n* = 323) (Table 4). Older age (51 ± 16 years); a higher percentage of cirrhosis and HCC; lower AST, ALT, total bilirubin, INR, AFP, HBV DNA, FIB-4, and platelets 12M; and higher AST 12M, ALT 12M, AFP 12M, and FIB-4 12M were significantly associated with the subgroup with △FIB-4 ≥ 0 (Table 4). Further analyses of the noncirrhotic and cirrhotic patients revealed that lower AST, ALT, total bilirubin, INR, AFP, HBV DNA, FIB-4, and platelets 12M; and higher platelets, AST 12M, ALT 12M, and FIB-4 12M were significantly associated with the subgroup with △FIB-4 ≥ 0 (Appendix A). In all (*n* = 105) or cirrhotic patients (*n* = 93) with HCC, lower AST, ALT, total bilirubin, AFP, FIB-4, and platelets 12M; and higher AST 12M and FIB-4 12M were significantly associated with the subgroup with △FIB-4 ≥ 0 (Appendix A). In noncirrhotic patients with HCC (*n* = 12), only lower AFP 12M was significantly associated with the subgroup with △FIB-4 ≥ 0, perhaps because of small number of cases (Appendix A).

### 2.6. Cumulative Incidence of HCC Stratified by the Combination of On-Treatment Changes in FIB-4 and 1-Year FIB-4 Values

For the entire cohort, a combination of FIB-4 12M and △FIB-4 could stratify the cumulative risk of HCC into four subgroups (*p* < 0.0001 by the log-rank test). Among them, 135 with FIB-4 12M ≥ 2.56 and △FIB-4 ≥ 0 exhibited the highest risk of HCC (HR: 25.58, 95% CI: 13.31–49.15, *p* < 0.0001), and 795 with FIB-4 12M < 2.56 and △FIB-4 < 0 exhibited the lowest risk of HCC (HR: 1 as reference) (Table 5, Figure 2A. A combination of FIB-4 12M and △FIB-4 was used to stratify the cumulative risk of HCC into three subgroups in the noncirrhotic and cirrhotic subgroups (each subgroup *p* < 0.0001 by the log-rank test). Among the noncirrhotic patients, 88 with FIB-4 12M ≥ 1.58 and △FIB-4 ≥ 0 exhibited the highest risk of HCC (HR: 40.35, 95% CI: 5.107–318.7, *p* < 0.0001), and 459 with FIB-4 12M < 1.58 and △FIB-4 < 0 exhibited the lowest risk of HCC (HR: 1 as reference) (Table 5, Figure 2B). Among the cirrhotic patients, 89 with FIB-4 12M ≥ 2.88 and △FIB-4 ≥ 0 exhibited the highest risk of HCC (HR: 9.576, 95% CI: 5.033–18.22, *p* < 0.0001), and 203 with FIB-4 12M < 2.88 and △FIB-4 < 0 exhibited the lowest risk of HCC (HR: 1 as reference) (Table 5, Figure 2C).

We summarise the predictive algorithm for HCC risk in the patients with CHB treated with entecavir according to FIB-4 12M (optimal cutoff value of 2.56 for the entire cohort) and △FIB-4 (Figure 3A) or their baseline liver cirrhosis status, FIB-4 12M (optimal cutoff values of 1.58 for the noncirrhotic subgroup and 2.88 for the cirrhotic subgroup), and △FIB-4 (Figure 3B). Regardless of baseline liver cirrhosis status, CHB patients who achieved FIB-4 12M < 2.56/△FIB-4 < 0 exhibited cumulative 3- and 5-year HCC risks of 1.1% and 1.7%, respectively, and those who achieved FIB-4 12M ≥ 2.56/△FIB-4 ≥ 0 exhibited the highest 3- and 5-year HCC risks of 15.2% and 35.3%, respectively (Figure 3A). The noncirrhotic patients who achieved FIB-4 12M < 1.58 or FIB-4 12M ≥ 1.58/△FIB-4 < 0 exhibited cumulative 3- and 5-year HCC risks of 0.3% and 0.6%, respectively. The cirrhotic patients who achieved FIB-4 12M ≥ 2.88/△FIB-4 ≥ 0 exhibited the highest 3- and 5-year HCC risks of 20.9% and 44.4%, respectively (Figure 3B).

### 2.7. Performance of the Predictive Algorithm for HCC in the Cirrhotic Subgroup with Liver Biopsies

Because liver cirrhosis was only verified by histology in 122 patients, we attempted to derive the optimal cutoffs for APRI, APRI 12M, FIB-4, and FIB-4 12M and determine their AUROCs, sensitivity, specificity, PPV, NPV, and accuracy in predicting HCC risk (Appendix A). Although they were significantly associated with HCC according to univariate Cox regression analysis, they could not coexist in the same multivariate Cox regression model (Appendix A). Therefore, we validated the performance of the predictive algorithm in this subgroup of patients. The AUROCs of the various FIB-4-based and APRI-based algorithms for predicting HCC risk in all cirrhotic patients and the cirrhotic subgroup with liver biopsies were comparable (Appendix A). The AUROCs of APRI 12M and FIB-4 12M for predicting HCC risk in the subgroup with liver biopsies were numerically higher than those in all cirrhotic patients (Appendix A). The AUROCs of FIB-4 12M for predicting HCC risk were numerally higher than those of APRI 12M and significantly higher than those of FIB-4 and APRI in all cirrhotic patients and the subgroup with liver biopsies (Appendix A). Similarly, a combination of FIB-4 12M and △FIB-4 could stratify the cumulative risk of HCC into three subgroups in the cirrhotic subgroup with liver biopsies (*p* < 0.001 by the log-rank test) (Appendix A).

### 2.8. Performance of the Predictive Algorithm in Cohorts Who Received 3 or 5 Years of Continuous Entecavir Therapy

To further validate the performance of the predictive algorithm in patients who had received entecavir therapy for a longer period, we selected the cohorts of patients who had received ≥ 3 (*n* = 1053) and ≥ 5 years (*n* = 472) of continuous entecavir therapy for further analysis. The AUROCs of the various FIB-4-based and APRI-based algorithms for predicting HCC risk in all patients, the 3-year cohort, or the 5-year cohort, in either the noncirrhotic or cirrhotic subgroup, were comparable (Appendix A). The sensitivity, specificity, PPV, NPV, and accuracy according to the cutoff values for APRI and FIB-4 indices at baseline or after 1 year of treatment in predicting HCC risk in these two cohorts are shown in Appendix A.

In the patients who received ≥ 3 years of continuous entecavir therapy, a combination of FIB-4 12M and △FIB-4 could stratify the cumulative risk of HCC into four subgroups in all patients (*p* < 0.0001 by the log-rank test), or into three subgroups in the noncirrhotic or cirrhotic subgroup (each subgroup *p* < 0.0001 by the log-rank test) (Appendix A). These results are in line with those for the entire cohort (Figure 2).

### 2.9. Comparison of AUROCs among Different HCC Prediction Models

Until now, some simple HCC risk prediction models for treatment-naïve CHB patients or patients under long-term NA therapy have been proposed, such as PAGE-B, REACH-B, CU-HCC, APA-B, CAGE-B, and SAGE-B scores [19,20,21,22,23,24,25,26]. We determined the C-statistic and time-dependent AUROCs of the proposed FIB-4-based predictive algorithm and the PAGE-B, REACH-B, CU-HCC, and APA-B scores, which we were able to calculate for predicting HCC risk after 2–5 years of entecavir therapy (Table 6). The APA-B score and FIB-4-based model had significantly higher predictive performance in predicting HCC risk compared to the other scores (Table 6, Appendix A). The APA-B score and FIB-4-based model had similar performance except after 3 years of entecavir therapy.

We further explored the ability of each model to identify patients with low-risk HCC. We calculated the cumulative incidences of HCC at 2 to 5 years of entecavir therapy among the low-risk group of patients in the present cohort according to different prediction models (Appendix A). The cumulative incidences of HCC at 5 years of therapy were 2.0%, 0%, 1.9%, and 2.6% among the low-risk group of patients defined by the PAGE-B, REACH-B, CU-HCC, and APA-B scores, respectively. The corresponding incidence in the low-risk noncirrhotic patients identified by the FIB-4-based model was 0.6%, which compared favorably with that from the other models. Although the REACH-B model identified a subgroup of patients whose 5-year HCC risk was 0%, the number of eligible patients was only 78, compared to 756 for the FIB-4-based model. Thus, the present predictive algorithm was able to identify a large number of patients with the lowest risk of HCC during entecavir therapy.

## 3. Discussion

A total of 105 (7.9%) patients in the present cohort developed HCC during a median follow-up of 4.1 years with a 4-year cumulative incidence of 7.2%. The cumulative incidences of HCC after 5 years of treatment were 9.9% (noncirrhotic: 2.4% and cirrhotic: 19.3%). A Taiwanese multicentre study in patients with predominantly compensated cirrhosis (90% Child-Pugh class A) reported a cumulative incidence of 11.3% after 5 years of entecavir treatment [27]. Another study in Caucasian CHB patients treated with entecavir or tenofovir for a median duration of 39 months revealed that the cumulative incidences of HCC after 5 years of treatment were 3.7% and 17.5% in the noncirrhotic and cirrhotic subgroups, respectively [28]. The incidence of HCC in untreated patients with CHB is generally believed to be higher in Asians compared to in Caucasians [29]. The factors of older age, male sex, an increased severity of liver fibrosis (e.g., a liver stiffness measurement above 12 kPa), a lower platelet count, and DM are associated with a higher risk of HCC in patients with CHB receiving long-term NA therapy [19,20,21,24,25,26,27,28,30]. However, the severity of liver fibrosis may be variable and subject to measurement error owing to the lack of a standardised method. This may have, in part, accounted for the varying incidences of HCC across studies [27,28,29]. Several risk scores have been proposed on the basis of these risk factors to predict HCC [19,21,24,25,26]. Obviously, liver fibrosis status is an integral component of these scores. Although current evidence suggests that elastography exhibits superior diagnostic performance to noninvasive fibrosis indices in the assessment of liver fibrosis [31], given their simplicity and availability, noninvasive fibrosis indices may be used as a surrogate marker of the severity of liver fibrosis to gauge the risk of HCC in clinical practice.

The FIB-4 index is a well-known noninvasive index that evaluates liver fibrosis in patients with CHB. Recently, the relationship between on-treatment FIB-4 values during entecavir treatment and HCC incidence has been investigated. Tada et al. reported that the FIB-4 after 24 weeks of therapy had adequate predictive performance for HCC incidence in patients with CHB according to a time-dependent ROC analysis [32]. An FIB-4 value ≥ 2.65 at 24 weeks of therapy was a risk factor for developing HCC with an HR of 5.03 (*p* < 0.001) [32]. We previously demonstrated that the FIB-4 after 1 year of treatment exhibited higher predictive performance for HCC compared with baseline values and that the FIB-4 at 1 year of treatment was an independent predictor of HCC, cirrhotic events, and mortality in compensated cirrhotic patients [33]. Furthermore, we demonstrated that cirrhotic patients with a decline in FIB-4 from baseline to after 1 year of treatment exhibited a significantly lower risk of HCC than did those with an increased FIB-4 [33].

In the present study, we systemically explored the predictive performance of absolute noninvasive fibrosis index values at baseline and after 1 year of treatment and observed changes between these two time points to determine HCC risk in patients with CHB receiving long-term entecavir therapy. We demonstrated that diabetes mellitus, AFP 12M, FIB-4 12M, and changes in FIB-4 values were independent predictors of HCC for the entire cohort. The APRI and FIB-4 values after 1 year of treatment exhibited higher predictive performance for HCC compared with those at baseline (Appendix A). Dynamic change in FIB-4 was also an HCC risk predictor. A combination of FIB-4 12M and △FIB-4 provides a convenient algorithm predictive of HCC risk in patients with CHB on long-term entecavir therapy. Regardless of baseline liver fibrosis status, the simple algorithm was able to stratify all patients into four subgroups according to the risk of HCC. However, if the cirrhosis status of patients could be ascertained at baseline, the alternative algorithm might provide a more precise estimate of future HCC risk. Of note, the cirrhotic patients with FIB-4 12M ≥ 2.88 and △FIB-4 ≥ 0 exhibited a very high risk of HCC (20.9% at 3 years and 44.4% at 5 years). By contrast, the noncirrhotic patients with FIB-4 12M < 1.58 or FIB-4 12M ≥ 1.58 and △FIB-4 < 0 exhibited a very low risk of HCC (0.3% at 3 years and 0.6% at 5 years) (Figure 3). Our findings may have implications regarding the prediction of HCC risk in patients with CHB on long-term NA therapy using noninvasive fibrosis indices. First, the FIB-4 index after 1 year of treatment may obviate the confounding effect of necroinflammation on the fibrosis measurement and thus better reflect the actual extent of liver fibrosis and predict HCC risk than that at baseline. Second, a decline in FIB-4 values during the first year of entecavir therapy (75.6% in the present study) may imply ongoing fibrosis regression in addition to the resolution of necroinflammation and thus represent a favorable predictor for future HCC risk. Instead, an increase in FIB-4 values at 1 year of treatment may indicate fibrosis progression despite NA therapy. The observation that patients with a decline or increase in FIB-4 values during the first year of therapy exhibited an increase or decrease in platelet count at 1 year, respectively, compared to the baseline level, supports this speculation (Table 4). This notion is in agreement with our observation that the noncirrhotic patients with FIB-4 12M ≥ 1.58 and cirrhotic patients with FIB-4 12M ≥ or <2.88 who exhibited an increase in FIB-4 at 1 year of entecavir therapy had a significantly higher risk of HCC compared with their counterparts who had a decline in FIB-4 at 1 year. The present study provides evidence in support of the relevant role that FIB-4 kinetics during the first year of entecavir therapy may play in predicting future HCC risk. Further research is warranted to elucidate the mechanisms underlying the correlation between on-treatment FIB-4 values and concomitant histological changes.

Our findings may have some clinical implications. First, this was a cohort study with a large patient number that aimed to stratify HCC risk by combining the FIB-4 index and its dynamic change during the first year of NA therapy for the development of a predictive algorithm for clinical application. Second, our simple and inexpensive predictive algorithm exhibited performance comparable to that of the APA-B score but better than that of the PAGE-B, REACH-B, and CU-HCC scores and might help identify a subgroup of noncirrhotic patients with a minimal risk of HCC (0.6% at 5 years), which is lower than those of the low-risk groups identified by the current available risk models and also lower than the threshold for implementing the HCC surveillance (0.2% per year) recommended by the AASLD [34,35]. Future efforts may focus on the implementation of additional biomarkers to further identify patients with zero risk of HCC, for whom HCC surveillance might no longer be needed or its interval could be extended. Third, our predictive algorithm likewise helps to identify a subgroup of cirrhotic patients with a very high risk of HCC (44.4% at 5 years), who may require an intensive HCC surveillance program and who represent the optimal candidates for HCC chemoprevention therapy if it becomes available in the near future.

There are also some limitations to note. First, this was a retrospective analysis with a median follow-up period of 4.1 years and only 105 cases of HCC (*n* = 12 in the noncirrhotic patient subgroup). A larger number of patients with a longer follow-up period is required to demonstrate the predictive performance of this algorithm for late HCC development during long-term NA therapy. Second, although we validated our findings by conducting subgroup analyses in cirrhotic patients with liver biopsies and patients receiving 3 or 5 years of continuous entecavir therapy, another cohort of patients is required for the external validation of the predictive performance of the cutoffs we proposed for the FIB-4 index and its change at 1 year of entecavir therapy. The optimal time point for assessing the predictive role of on-treatment FIB-4 changes for the assessment of HCC risk must be further investigated. Third, the severity of liver fibrosis was only assessed using noninvasive fibrosis indices, and the longitudinal changes in these indices during entecavir therapy were not corroborated by elastography or even histological examination. This was because elastography was not an option for us for the majority of the study period and many of the enrolled patients did not provide consent for percutaneous liver biopsy. The implementation of elastography is warranted to improve the performance of the predictive model in the future.

## 4. Materials and Methods 

### 4.1. Patient Recruitment and Definitions

Our study was a real-world cohort study with retrospective analysis from two tertiary care medical centres in Taiwan. A total of 1325 NA-naïve patients with CHB who received entecavir monotherapy (0.5 mg once daily) at China Medical University Hospital (*n* = 596) in Taichung and Kaohsiung Chang Gung Memorial Hospital (*n* = 729) in Kaohsiung were enrolled from January 2007 until August 2012. The inclusion criteria for this cohort included being seropositive for HBsAg for more than 6 months, having an HBV viral load ≥2000 IU/mL at recruitment, and an entecavir treatment duration of more than 12 months. We excluded patients who were NA-experienced (*n* = 553); had an HBV viral load < 2000 IU/mL at enrollment (*n* = 75); had HCC at baseline or developed HCC within 1 year of treatment (*n* = 318); had decompensated cirrhosis (*n* = 49); had evidence of autoimmune hepatitis, alcoholic liver disease, or viral coinfections (*n* = 71); had received immunosuppressive therapy (*n* = 317); or had a treatment duration < 12 months (*n* = 574) [24].

Baseline and on-treatment characteristics and laboratory parameters were collected every 3–6 months during treatment, including age, sex, cirrhosis status, diagnosis of DM, albumin, AST, ALT, total bilirubin, INR, platelet count, AFP, and HBV DNA level.

Noninvasive fibrosis indices, including APRI and FIB-4, were calculated according to the following formulas [13,14]:APRI = ((AST [/ULN])/(Platelet count [109/L])) × 100(1)

Note: The upper limit of normal for AST was 30 U/L.
FIB–4 = (Age [years] × AST [U⁄L])/(Platelet count [10^9^⁄L] × ALT [U⁄L]^1/2^) (2)

In addition to the baseline and on-treatment (1 year) index values, the on-treatment changes in index, defined as the index value at 1 year minus the index value at baseline, were calculated. The on-treatment change in index was presented as index (12M−0M) or △index. We defined VR as a serum HBV DNA level of <50 IU/mL during entecavir therapy [24]. Abdominal ultrasonography and AFP were performed every 3–6 months during treatment for HCC surveillance as per the National Health Insurance guidelines of Taiwan. Baseline cirrhosis status was confirmed either by liver biopsy (*n* = 122) or abdominal ultrasonography (*n* = 359), which showed consistent findings suggestive of cirrhosis and clinical manifestations, including splenomegaly, gastroesophageal varices, ascites, or thrombocytopenia [36]. HCC was diagnosed with either multiphasic computed tomography (CT) or multiphasic magnetic resonance imaging (MRI) and/or histology according to the practice guidelines of the European Association for the Study of the Liver (EASL) and American Association for the Study of Liver Diseases (AASLD) [34,35]. The presence of DM was ascertained through (1) an HbA1c measurement ≥ 6.5 percent, (2) a fasting glucose test result ≥ 126 mg per dL, (3) results of two random glucose tests ≥ 200 mg per dL with classic symptoms of hyperglycemia or a hyperglycemic crisis, (4) a two-hour glucose test result ≥200 mg per dL during an oral glucose tolerance test, or (5) a medical record of anti-diabetic drug use [37].

This study was conducted in accordance with the 1975 Declaration of Helsinki. All patients provided written informed consent, and the study was approved by the research ethics committees of China Medical University Hospital and Chang Gung Memorial Hospital.

### 4.2. Statistical Analysis

For descriptive data, we analysed the categorical variables between the two groups by using the chi-square or Fisher exact test, as applicable. Continuous variables were assessed for normality of distribution by using the Kolmogorov–Smirnov test, and they were non-normally distributed. We used the Mann–Whitney U test to compare the continuous variables, which are expressed as the median ± interquartile range among the study subgroups. Additionally, we used the receiver operating characteristic (ROC) curve and Youden index to identify the optimal cutoff values for the noninvasive fibrosis indices to differentiate HCC risk. We calculated and listed the sensitivity, specificity, positive predictive value, negative predictive value, and accuracy according to the optimal cutoff values for the APRI and FIB-4 indices (Appendix A). We compared the area under the receiver operating characteristics (AUROCs) of APRI and FIB-4 at the different time points to predict HCC risk in CHB patients with or without cirrhosis using the DeLong test (Appendix A). The comparison of △APRI (12M-0M) and △FIB-4 (12M–0M) to predict HCC risk by AUROC was also performed in different subgroups (Appendix A). Finally, we chose the FIB-4-based model for further analysis because of the following reasons. 1) Both FIB-4 12M and APRI 12M had acceptable AUROCs for predicting HCC in cirrhotic patients and excellent AUROCs in noncirrhotic and all patients. The AUROC of FIB-4 12M was numerically higher than that of APRI 12M (Appendix A). 2) The predictive performance of △FIB-4 (12M-0M) for HCC was significantly higher than that of △APRI (12M–0M) for cirrhotic patients and was numerically higher than that of △APRI (12M-0M) for noncirrhotic patients (Appendix A). 3) FIB-4 and APRI were used as covariates and were confounding factors according to multivariate Cox regression analysis. Kaplan–Meier analysis with the log-rank test was performed to compare the cumulative HCC incidences among the different subgroups stratified according to FIB-4 12M or △FIB-4 (12M–0M) or the combination of FIB-4 12M and △FIB-4 (12M–0M). The hazard ratios (HRs) for HCC risk predictors were accessed by univariate and multivariate Cox regression analyses. Stepwise multivariate analysis was performed with the variables with a *p* value of less than 0.25 in univariate analysis. Multiple imputation was used to deal with missing data for the analysed variables (Supplementary Methods and Appendix A) [38,39]. Statistical analyses were performed using SPSS Version 21.0 (Armonk, NY: IBM Corp.) and STATA Version 16. A two-tailed *p* value of less than 0.05 was considered statistically significant.

## 5. Conclusions

On-treatment changes in FIB-4 and 1-year FIB-4 values are independent predictors of HCC in patients with CHB receiving long-term entecavir therapy. A combination of these two predictors may help identify patients with noncirrhotic CHB with the lowest risk of HCC during entecavir therapy.

## Figures and Tables

**Figure 1 cancers-12-01177-f001:**
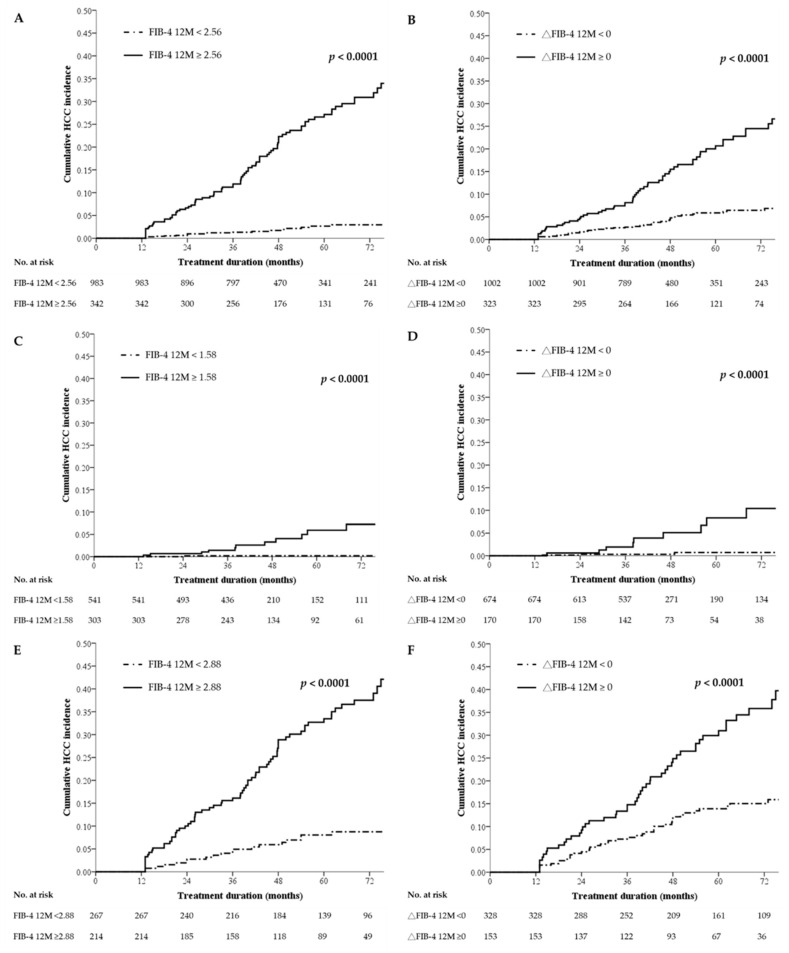
HCC risk stratification by FIB-4 12M or △FIB-4 in treatment-naïve patients with CHB: all patients (**A**) and (**B**), noncirrhotic patients (**C**) and (**D**), and cirrhotic patients (**E**) and (**F**). CHB—chronic hepatitis B; FIB-4—fibrosis index based on four factors; HCC—hepatocellular carcinoma; M—months.

**Figure 2 cancers-12-01177-f002:**
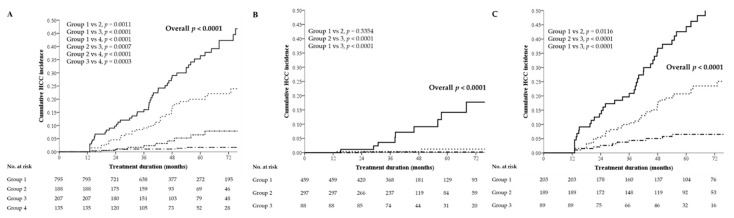
HCC risk stratification by a combination of FIB-4 12M and △FIB-4. (**A**) All patients: 
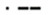
 Group 1, FIB-4 12M < 2.56 and △FIB-4 < 0; 
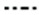
 Group 2, FIB-4 12M < 2.56 and △FIB-4 ≥ 0; 
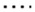
 Group 3, FIB-4 12M ≥ 2.56 and △FIB-4 < 0; 

 Group 4, FIB-4 12M ≥ 2.56 and △FIB-4 ≥ 0. (**B**) Noncirrhotic patients: 
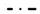
 Group 1, FIB-4 12M < 1.58 and △FIB-4 < 0; 
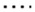
 Group 2, FIB-4 12M ≥ 1.58 and △FIB-4 < 0 or FIB-4 12M < 1.58 and △FIB-4 ≥ 0; 

 Group 3, FIB-4 12M ≥ 1.58 and △FIB-4 ≥ 0. (**C**) Cirrhotic patients: 
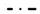
 Group 1, FIB-4 12M < 2.88 and △FIB-4 < 0; 
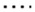
 Group 2 FIB-4 12M ≥ 2.88 and △FIB-4 < 0 or FIB-4 12M < 2.88 and △FIB-4 ≥ 0; 

 Group 3 FIB-4 12M ≥ 2.88 and △FIB-4 ≥ 0. FIB-4—fibrosis index based on four factors; HCC—hepatocellular carcinoma; M—months.

**Figure 3 cancers-12-01177-f003:**
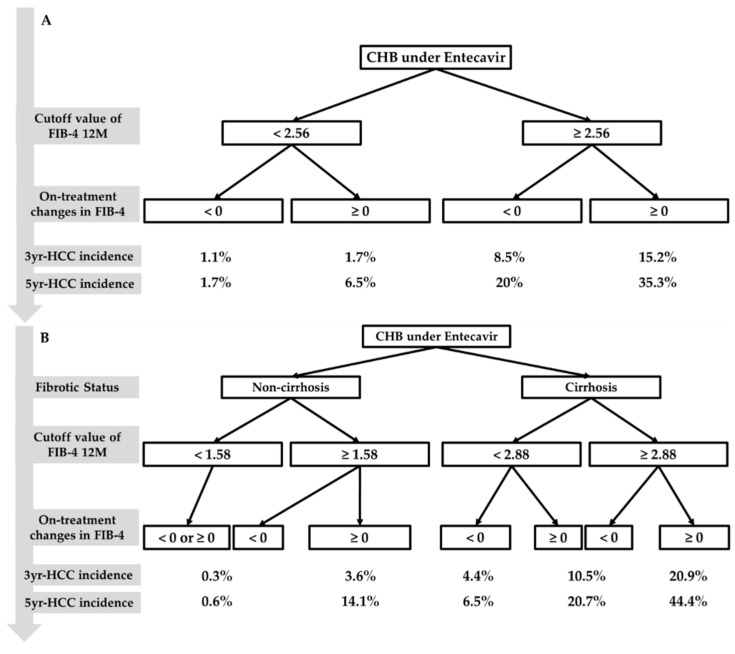
Algorithm for the prediction of HCC risk in CHB patients (**A**) without stratification by baseline liver fibrosis status and (**B**) stratified by baseline liver fibrosis status. CHB—chronic hepatitis B; FIB-4—fibrosis index based on four factors; HCC—hepatocellular carcinoma; M—months; yr—years.

**Table 1 cancers-12-01177-t001:** Baseline and on-treatment characteristics of treatment-naïve patients with chronic hepatitis B (CHB) (*n* = 1325).

Variables Median ± IQR or *n* (%)	Total *n* = 1325	HCC Occurrence		*p* Value
		Yes (*n* =105)	No (*n* = 1220)	
Age (years)	50 ± 17	58 ± 14	49 ± 17	<0.0001
Sex (male)	963 (72.7)	79 (75.2)	884 (72.5)	0.5397
HBeAg-positive status	475 (35.8)	29 (27.6)	446 (36.6)	0.0669
Cirrhosis status	481 (36.3)	93 (88.6)	388 (31.8)	<0.0001
Diabetes mellitus (yes)	158 (11.9)	22 (21.0)	136 (11.1)	0.0029
Albumin, g/dL	4.11 ± 0.6	3.9 ± 0.7	4.2 ± 0.6	<0.0001
AST, U/L	72 ± 115	60 ± 46	73 ± 123	0.0028
ALT, U/L	106 ± 207	64 ± 59	113 ± 220	<0.0001
Total bilirubin, mg/dL	1 ± 0.78	1.1 ± 0.7	1 ± 0.8	0.5459
INR	1.08 ± 0.15	1.11 ± 0.17	1.07 ± 0.14	0.0090
Platelets, ×10^3^/μL	163 ± 76	120 ± 70	167 ± 73.5	<0.0001
Histology grading				
F 1/2/3/4/NA	57/84/30/122/1032	0/2/1/17/85	57/82/29/105/947	
A 0/1/2/3/NA	40/138/64/51/1032	3/11/2/4/85	37/127/62/47/947	
AFP, ng/mL	6.01 ± 10.31	8.25 ± 11.94	5.82 ± 10.18	0.0002
AFP 12M, ng/mL	3.41 ± 2.35	5.63 ± 4.63	3.28 ± 2.09	<0.0001
HBV DNA, log IU/mL	5.96 ± 7.28	5.49 ± 6.46	6.04 ± 7.34	<0.0001
Time to VR (M)	6.0 (6.3)	6.0 (0)	6.0 (6.4)	0.9267
VR 12M (yes)	1160 (87.5)	98 (93.3)	1062 (87)	0.0613
APRI	1.22 ± 2.1	1.43 ± 1.73	1.19 ± 2.13	0.1881
APRI (≥0.90)	855 (64.5)	80 (76.2)	775 (63.5)	0.0092
APRI 12M	0.42 ± 0.37	1.04 ± 0.67	0.41 ± 0.31	<0.0001
APRI 12M (≥0.53)	473 (35.7)	92 (87.6)	381 (31.2)	<0.0001
△APRI (12M-0M) (≥0)	136 (10.3)	33 (31.4)	103 (8.4)	<0.0001
FIB-4	2.45 ± 2.69	4.09 ± 4.03	2.37 ± 2.51	<0.0001
FIB-4 (≥2.53)	637 (48.1)	82 (78.1)	555 (45.5)	<0.0001
FIB-4 12M	1.64 ± 1.56	4.09 ± 3.22	1.55 ± 1.38	<0.0001
FIB-4 12M (≥2.56)	342 (25.8)	84 (80.0)	258 (21.1)	<0.0001
△FIB-4 (12M–0M) (≥0)	323 (24.4)	60 (57.1)	263 (21.6)	<0.0001
Treatment duration (year)	4.09 (2.94)	3.29 (2.32)	4.11 (3.00)	<0.0001
Follow-up period (year)	4.09 (2.94)	3.29 (2.32)	4.11 (3.00)	<0.0001

Abbreviations: AFP—alpha-fetoprotein; ALT—alanine aminotransferase; AST—aspartate aminotransferase; APRI—AST/PLT ratio index; CHB—chronic hepatitis B; DNA—deoxyribonucleic acid; FIB-4—fibrosis index based on four factors; HBeAg—hepatitis B e antigen; HBV—hepatitis B virus; HCC—hepatocellular carcinoma; INR—international normalised ratio; IQR—interquartile range; M—month; NA—not available; PLT—platelet; VR—virological response.

**Table 2 cancers-12-01177-t002:** Univariate Cox regression analysis of risk factors associated with HCC (*n* = 105) in all patients (*n* = 1325).

Variables Median ± IQR or *n* (%)	Univariate	
	Hazard Ratio (95% CI)	*p* Value
Age (year)	1.056 (1.039–1.074)	<0.0001
Sex, male vs. female	1.103 (0.708–1.718)	0.6653
HBeAg, positive vs. negative	0.717 (0.467–1.100)	0.1272
Cirrhosis status, yes vs. no	11.29 (6.180–20.63)	<0.0001
Diabetes mellitus, yes vs. no	2.280 (1.424–3.650)	0.0006
Albumin, g/dL	0.406 (0.294–0.559)	<0.0001
AST, U/L	0.998 (0.997–1.000)	0.0117
ALT, U/L	0.998 (0.996–0.999)	0.0006
Total bilirubin, mg/dL	0.929 (0.829–1.042)	0.2087
INR	1.618 (0.866–3.021)	0.1311
Platelets, ×10^3^/μL	0.985 (0.982–0.989)	<0.0001
AFP, ng/mL	0.999 (0.996–1.001)	0.3951
AFP 12M, ng/mL	1.005 (1.002–1.007)	<0.0001
HBV DNA, log IU/mL	1.000 (1.000–1.000)	0.1854
VR 12M, yes vs. no	1.931 (0.897–4.159)	0.0925
APRI, ≥0.90 vs. <0.90	1.991 (1.270–3.121)	0.0027
APRI 12M, ≥0.53 vs. <0.53	11.79 (6.596–21.09)	<0.0001
△APRI (12M-0M), ≥0 vs. <0	3.845 (2.546–5.807)	<0.0001
FIB-4, ≥2.53 vs. <2.53	3.919 (2.467–6.223)	<0.0001
FIB-4 12M, ≥2.56 vs. <2.56	11.31 (7.009–18.25)	<0.0001
△FIB-4 (12M–0M), ≥0 vs. <0	3.953 (2.685–5.819)	<0.0001

Abbreviations: AFP—alpha-fetoprotein; ALT—alanine aminotransferase; AST—aspartate aminotransferase; APRI—AST/PLT ratio index; CHB—chronic hepatitis B; CI—confidence interval; DNA—deoxyribonucleic acid; FIB-4—fibrosis index based on four factors; HBeAg—hepatitis B e antigen; HBV—hepatitis B virus; HCC—hepatocellular carcinoma; INR—international normalised ratio; IQR—interquartile range; M—month; PLT—platelet; VR—virological response.

**Table 3 cancers-12-01177-t003:** Multivariate Cox regression analysis of risk factors (FIB-4 based model) associated with HCC in patients with noncirrhotic and cirrhotic CHB.

Risk Factors	Multivariate	
	Hazard Ratio (95% CI)	*p* Value
**All patients (*n* = 1325)**		
Diabetes mellitus, yes vs. no	1.726 (1.076–2.770)	0.0235
AFP 12M, ng/mL	1.005 (1.003–1.008)	<0.0001
FIB-4 12M, ≥2.56 vs. <2.56	9.198 (5.610–15.08)	<0.0001
△FIB-4 (12M-0M), ≥0 vs. <0	2.353 (1.585–3.495)	<0.0001
**Noncirrhotic patients (*n* = 844)**		
FIB-4 12M, ≥1.58 vs. <1.58	12.10 (1.531–95.60)	0.0181
△FIB-4 (12M-0M), ≥0 vs. <0	7.013 (1.874–26.24)	0.0038
**Cirrhotic patients (*n* = 481)**		
Sex, male vs. female	1.758 (1.082–2.856)	0.0226
Diabetes mellitus, yes	1.665 (1.006–2.756)	0.0472
AFP 12M, ng/mL	1.008 (1.005–1.012)	<0.0001
FIB-4 12M, ≥2.88 vs. <2.88	4.821 (2.908–7.992)	<0.0001
△FIB-4 (12M-0M), ≥0 vs. <0	1.981 (1.301–3.016)	0.0014

Abbreviations: AFP—alpha-fetoprotein; CHB—chronic hepatitis B; CI—confidence interval; FIB-4—fibrosis index based on four factors; HCC—hepatocellular carcinoma; M—month.

**Table 4 cancers-12-01177-t004:** Characteristics of treatment-naïve patients with CHB stratified by △FIB-4.

Variables Median ± IQR or *n* (%)	△FIB-4 < 0 *n* = 1002	△FIB-4 ≥ 0 *n* = 323	*p* Value
**Baseline**			
Age (years)	49 ± 17	51 ± 16	0.002
Sex (male)	727 (72.6)	236 (73.1)	0.719
HBeAg-positive status (yes)	367 (36.6)	108 (33.4)	0.298
Diabetes mellitus (yes)	120 (11.9)	38 (11.8)	0.919
Cirrhosis (yes)	328 (32.7)	153 (47.4)	<0.001
HCC	45 (4.5)	60 (18.6)	<0.001
Albumin, g/dL	4.1 ± 0.6	4.2 ± 0.5	0.163
AST, U/L	92 ± 187	46 ± 26	<0.001
ALT, U/L	137 ± 316	60 ± 50	<0.001
Total bilirubin, mg/dL	1.1 ± 1.0	0.9 ± 0.6	<0.001
INR	1.09 ± 0.16	1.06 ± 0.13	<0.001
Platelets, ×10^3^/μL	162 ± 75	166 ± 83	0.297
AFP, ng/mL	6.53 ± 13.3	4.87 ± 5.25	<0.001
HBV DNA, log IU/mL	6.19 ± 7.43	5.36 ± 6.54	<0.001
FIB-4	2.69 ± 3.00	1.73 ± 1.72	<0.001
**One-year treatment**			
AST 12M, U/L	26 ± 11	31 ± 16	<0.001
ALT 12M, U/L	26 ± 16	29 ± 18	<0.001
Platelets 12M, ×10^3^/μL	171 ± 77	141 ± 81	<0.001
AFP 12M, ng/mL	3.33 ± 2.25	3.69 ± 2.97	0.009
VR 12M (yes)	871 (86.9)	289 (89.5)	0.228
FIB-4 12M	1.53 ± 1.36	2.11 ± 2.34	<0.001

Abbreviations: AFP—alpha-fetoprotein; ALT—alanine aminotransferase; AST—aspartate aminotransferase; CHB—chronic hepatitis B; DNA—deoxyribonucleic acid; FIB-4—fibrosis index based on four factors; HBV—hepatitis B virus; HCC—hepatocellular carcinoma; INR—international normalised ratio; IQR—interquartile range; M—months; VR—virological response.

**Table 5 cancers-12-01177-t005:** HCC risk stratification by a combination of FIB-4 12M and △FIB-4 in patients with CHB.

Combined risk factors	Crude HR	95% CI	*p* Value
**All patients (*n* = 1325)**			
FIB-4 12M < 2.56 and △FIB-4 < 0 (*n* = 795)	1		
FIB-4 12M < 2.56 and △FIB-4 ≥ 0 (*n* = 188)	3.673	1.560–8.649	0.0029
FIB-4 12M ≥ 2.56 and △FIB-4 < 0 (*n* = 207)	11.74	5.948–23.18	<0.0001
FIB-4 12M ≥ 2.56 and △FIB-4 ≥ 0 (*n* = 135)	25.58	13.31–49.15	<0.0001
**Noncirrhotic patients (*n* = 844)**			
FIB-4 12M < 1.58 and △FIB-4 < 0 (*n* = 459)	1		
FIB-4 12M ≥ 1.58 and △FIB-4 < 0 or FIB-4 12M < 1.58 and △FIB-4 ≥ 0 (*n* = 297)	3.076	0.279–33.93	0.3589
FIB-4 12M ≥ 1.58 and △FIB-4 ≥ 0 (*n* = 88)	40.35	5.107–318.7	<0.0001
**Cirrhotic patients (*n* = 481)**			
FIB-4 12M < 2.88 and △FIB-4 < 0 (*n* = 203)	1		
FIB-4 12M ≥ 2.88 and △FIB-4 < 0 or FIB-4 12M < 2.88 and △FIB-4 ≥ 0 (*n* = 189)	3.625	1.897–6.927	<0.0001
FIB-4 12M ≥ 2.88 and △FIB-4 ≥ 0 (*n* = 89)	9.576	5.033–18.22	<0.0001

Abbreviations: CHB—chronic hepatitis B; CI—confidence interval; FIB-4—fibrosis index based on four factors; HCC—hepatocellular carcinoma; HR—hazard ratio; M—months.

**Table 6 cancers-12-01177-t006:** C-statistic and time-dependent AUROCs for predicting HCC risk by using different risk scores.

Risk Scores	PAGE-B	REACH-B	CU-HCC	APA-B	FIB-4-Based Model
	AUROC (95% CI)	AUROC (95% CI)	AUROC (95% CI)	AUROC (95% CI)	AUROC (95% CI)
**2 years**	0.7379 (0.6526–0.8233)	0.6673 (0.5801–0.7545)	0.7666 (0.7033–0.8298)	0.8815 (0.8236–0.9395)	0.8192 (0.7482–0.8902)
**3 years**	0.7415 (0.6741–0.8089)	0.6640 (0.5950–0.7329)	0.7771 (0.7210–0.8333)	0.8820 (0.8393–0.9247)	0.8359 (0.7858–0.8860)
**4 years**	0.7665 (0.7163–0.8167)	0.6679 (0.6112–0.7246)	0.7857 (0.7433–0.8280)	0.8910 (0.8591–0.9229)	0.8701 (0.8341–0.9060)
**5 years**	0.7471 (0.6980–0.7962)	0.6535 (0.5992–0.7078)	0.7809 (0.7394–0.8225)	0.8775 (0.8450–0.9100)	0.8659 (0.8321–0.8997)
**C-statistic**	0.7394 (0.6920–0.7868)	0.6551 (0.6033–0.7069)	0.7755 (0.7366–0.8145)	0.8825 (0.8532–0.9118)	0.8736 (0.8434–0.9038)

Abbreviations: AUROC—area under the receiver operating characteristics; CI—confidence interval; FIB-4—fibrosis index based on four factors; HCC—hepatocellular carcinoma.

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
