# Peer review of "On-Treatment Changes in FIB-4 and 1-Year FIB-4 Values Help Identify Patients with Chronic Hepatitis B Receiving Entecavir Therapy Who Have the Lowest Risk of Hepatocellular Carcinoma"

_cancers, 2020, doi:10.3390/cancers12051177_

Round 1
Reviewer 1 Report
The authors answered all my questions
Reviewer 2 Report
After rereading the manuscript, the comments and the authors' answers: very clearly I think that the authors have done an important work to respond to the 17 major comments and the 7 minor comments of the third reviewer. Their answers seem to me to be quite clear, honest and detailed. This proofreading work brings even more interest and value to this manuscript, which deserves to be published as it stands.
This manuscript is a resubmission of an earlier submission. The following is a list of the peer review reports and author responses from that submission.
Round 1
Reviewer 1 Report
This study by Wang et al investigates the role of on treatment changes in FIB-4 and 1-year FIB-4 values in HBV patients treated with entecavir to identify those with low or high risk to develop HCC.
This study is a retrospective study and the authors clearly note that their results need further validation with a well designed prospective study.
However the results of this study are of interest and provide data and an easy to calculate algorithm to identify HBV treated patients with low or high risk to develop HCC and the underlying clinical meaning
Comments to be provided and discussed
In this study population what is the cut off value of FIB-4 in patients with established cirrhosis ?
In cirrhotic patients who had an increase in FIB-4 score during treatment (high risk to develop HCC)
a) what proportion (%) had this increase b) what was the reason for that? how many had no response to treatment (need for HBV DNA) or confounding factors (presence of diabetes or other risk factors)
In non cirrhotic patients who developed HCC a) what was the FIB-4 value before treatment and 1 year after and what was the reason for FIB -4 increase at 1 year (no response to treament, other co-factos (diabetes /, obesity?
These informations have a clinical input
Reviewer 2 Report
Wang et al. reported that a combination of on treatment changes in FIB-4 and 1-year FIB-4 values helps to identify CHB patients receiving ETV with lowest risk of HCC. This is a well written and clear paper. The major strength of the study is the large number of patients, the long follow up and the significant rate of HCC development.
Major concerns
1) It is unclear why cirrhosis was not considered at univariate cox regression analysis of risk factors associated with HCC (Table 2 and text) and subsequently to multivariate, although HCC occurrence was higher among cirrhotic patients.
2) It would be useful to identify cut-offs for FIB-4 12M and delta FIB4 for the whole cohort (and add a new figure for the entire population for HCC risk stratification) and only subsequently to divide into cirrhosis and non-cirrhosis. This is because the non-invasive diagnosis of cirrhosis may not always be accurate and the same FIB-4 identifies the stage of liver disease.
3) I suggest to mention in the discussion the other scores (GAG-HCC, CU-HCC, REACH-B and PAGEB) for the prediction of HCC in CHB. Moreover, it would be useful to analyze the performance of PAGE B in this cohort and make a comparison in terms of accuracy in identifying the group of low risk.
4) I suggest to report the accuracy of the cut offs reporting the sensitivity, specificity, PPV and NPV in both cirrhotic and non cirrhotics but also in the whole population. This would be very useful to better understand which cut off has very high NPV identifying the group of patients at almost zero risk
Minor concerns
1) Page 2 line 63 ALT are higher among patients with HCC than without HCC …and not lower as reported
2) Are you sure that US scan and AFP were performed every 3-6 months in all patients?
Reviewer 3 Report
In this study, Wang et al tried to address a very relevant clinical question regarding the issue of HCC risk development in patients with CHB under entecavir treatment. Actually, they investigated the predictive performance of on-treatment changes in FIB-4 score and 1-year FIB-4 values for HCC risk in CHB patients. Not surprisingly, they found that a combination of on-treatment changes in FIB-4 and 1-year FIB-4 values help to identify non-cirrhotic CHB patients receiving entecavir with the lowest HCC risk. The manuscript could be of interest but too many major and minor concerns have been raised that need to be addressed.
Major concerns
- In general, the study is interesting but its inevitable retrospective nature without a standard protocol of liver stiffness measurements (LSM) determinations is probably the reason which raised many uncertainties (see below).
- Introduction; p.2; lines 1-2: Please add “…..occurrence remains although, in a recent large cohort of 1951 Caucasian patients with CHB under ETV or TDF, the 8-year survival was found similar to the general population”. (Papatheodoridis et al, J Hepatol 2018;68:1129-36).
- Introduction; p.2; 1st par.: Please cite and also comment on other recent large multicenter studies on the value of LSM determinations such as, Papatheodoridis et al, J Hepatol 2020, doi: 10.1016/j.jhep.2020.01.007 and Papatheodoridis et al, Hepatology 2017;66:1444-53).
- Results; p.2; 1st par.: As the vast majority of patients with cirrhosis did not have a liver biopsy (359 out of 481; 75%!!) a subgroup analysis with those who have a firm diagnosis of cirrhosis at the histological level should also be done. The absence of liver histology and/or LSM determinations is a major drawback of the study.
- Results; p.2; 1st par.: You have stated that almost 8% of patients developed HCC during a median follow-up of 4 years. At least for this reviewer this frequency seems quite high. Why? Is this common in Asian patients with CHB? Also the dose of entecavir should be clearly stated for cirrhotic and non-cirrhotic patients and if HCC development was potentially affected by different doses.
- Results; p.2; 2nd par.: How the optimal values of APRI, APRI 12M, FIB-4 and FIB-4 12M were defined?
- Table 1: If I did not miss anything, I do not understand why you compare HCC vs. non-HCC patients. The development of HCC during a median of 4 years means a wide range. For instance, you cannot include patients with HCC development after 1-year or 1.5-years of ETV treatment because probably HCC was present before ETV initiation! So, I would suggest including only patients receiving 3 or 5 years of continuous ETV administration in your analysis.
- Table 1: Some baseline characteristics that could affect your results are missing and should be added such as, disease duration, duration up to treatment response, route of transmission, histological grading (necroinflammatory activity), total follow-up and treatment duration.
- Discussion: Given the inherent limitations of the study, I would suggest presenting the whole discussion and especially the conclusion section in a less declamatory way. There are many too strong statements that need attention.
- Discussion: The recent PAGE-B, SAGE-B and CAGE-B scores which are very simple and reliable risk scores for HCC prediction and surveillance should also be defined and compared with your supposed score in an attempt to show whether FIB-4 score is superior or at least not inferior with the abovementioned scores.
- Discussion: p.8, lines 1-2 from the bottom and p. 9, first line: From my point of view this statement is not correct. As you also stated in p.9 regarding the limitations of the study, you do not investigate liver fibrosis by at least LSM determinations which is a very reliable tool for fibrosis assessment!! Therefore, I would suggest doing again the analysis after changing the period of the study (e.g. between 2010 and 2018) in order to include LSM in your analysis. CHB is quite frequent in China so, I do not think to be a problem to recruit patients with LSM data.
- Materials and Methods; p.9: At least this reviewer does not understand why you chose the predictive performance of APRI and FIB-4 at baseline and 1-year of treatment as this period seems too short for such a chronic liver disease like CHB.
- Materials and Methods; p.10: The diagnosis of HCC should be updated according to the recent EASL (J Hepatol 2018) and AASLD (Hepatology 2018) clinical practice guidelines.
- Statistical analysis: Although I am not a biostatistician, I do not understand why you chose only the Mann-Whitney U (MWU) test. You should check first for normality of the distribution of variables by the Kolmogorov-Smirnof test and then you should decide accordingly the appropriate test (MWU test or unpaired t-test which is much more powerful).
- Statistical analysis: Also I cannot understand why only FIB-4 determinations were of value. The APRI score has more or less similar determinants. Why APRI was not valuable. Your reasons are not very convincing. As PLTs count is an independent risk factor in three recent studies in Caucasian CHB patients your exclusion of APRI is a little bit surprising.
- Statistical analysis: We need further and detailed information concerning the imputation process which was used at least as supplementary material.
- Discussion; p. 11; conclusion section: The last sentence is not surprising. I think this is already known. Non-cirrhotic CHB patients have extremely low incidence rate for HCC development (if you are pretty sure that there is not misdiagnosed and/or underdiagnosed underlying cirrhosis). Or not? Which is the additional help for these not high risk patients for HCC development?
Minor concerns
- The abstract and in particular the conclusion should be changed accordingly.
- Tables 1 and 4: I noticed that HBV DNA was lower in HCC group and those with on treatment changes in FIB-4 ≥0. Why?
- The kind of surveillance for HCC development is not clearly indicated in CHB patients. This issue seems obligatory and should be added.
- Discussion; p.8; lines 1-2: Please add also “LSM determinations above 12 kPa” as another factor that is associated with higher risk of HCC.
- Suppl. Table 5: I am not sure that AUROCs of 0.672, 0686 and 0.605 are very good in general!
- Any reason why only entecavir was used?
- References: Apart from Asian studies new updated studies in Caucasian CHB patients should be added.